# Enhanced phosphorylation of PERK in primary cultured neurons as an autonomous neuronal response to prion infection

**Misaki Tanaka[1], Takeshi Yamasaki[1¤], Rie Hasebe[1¤], Akio Suzuki[1], Motohiro Horiuchi●[1,2]***

**1** Laboratory of Veterinary Hygiene, Faculty of Veterinary Medicine, Hokkaido University, Sapporo, Japan,
**2** Global Station for Zoonosis Control, Global Institute for Collaborative Research and Education, Hokkaido University, Sapporo, Japan

¤ Current address: Institute for Genetic Medicine, Hokkaido University, Sapporo, Japan
* horiuchi@vetmed.hokudai.ac.jp

**Data Availability Statement:** All relevant data are within the manuscript and its Supporting Information files.

## Abstract

Conversion of cellular prion protein (PrP$^C$) into the pathogenic isoform of prion protein (PrP$^{Sc}$) in neurons is one of the key pathophysiological events in prion diseases. However, the molecular mechanism of neurodegeneration in prion diseases has yet to be fully elucidated because of a lack of suitable experimental models for analyzing neuron-autonomous responses to prion infection. In the present study, we used neuron-enriched primary cultures of cortical and thalamic mouse neurons to analyze autonomous neuronal responses to prion infection. PrP$^{Sc}$ levels in neurons increased over the time after prion infection; however, no obvious neuronal losses or neurite alterations were observed. Interestingly, a finer analysis of individual neurons co-stained with PrP$^{Sc}$ and phosphorylated protein kinase RNA-activated-like endoplasmic reticulum (ER) kinase (p-PERK), the early cellular response of the PERK-eukaryotic initiation factor 2 (eIF2α) pathway, demonstrated a positive correlation between the number of PrP$^{Sc}$ granular stains and p-PERK granular stains, in cortical neurons at 21 dpi. Although the phosphorylation of PERK was enhanced in prion-infected cortical neurons, there was no sign of subsequent translational repression of synaptic protein synthesis or activations of downstream unfolded protein response (UPR) in the PERK-eIF2α pathway. These results suggest that PrP$^{Sc}$ production in neurons induces ER stress in a neuron-autonomous manner; however, it does not fully activate UPR in prion-infected neurons. Our findings provide insights into the autonomous neuronal responses to prion propagation and the involvement of neuron-non-autonomous factor(s) in the mechanisms of neurodegeneration in prion diseases.

## Introduction

Prion diseases are a group of fatal neurodegenerative disorders in animals and humans, such as scrapie in sheep and goats, bovine spongiform encephalopathy, chronic wasting disease in cervids, and Creutzfeldt-Jakob disease (CJD) in humans. The presence of misfolded prion

**Funding:** This work was supported by Grant-in-aid for JSPS Research Fellow Grant Number JP18J12535 (M.T.). This work was also supported by a Grant-in-Aid for Science Research (A) (JSPS KAKENHI Grant Number JP 15H02475) (M.H.) and a grant from the Program for Leading Graduate Schools (F01) (M.H.) from the Ministry of Education, Culture, Sports, Science, and Technology, Japan. This work was also supported by grants for TSE research (H29-Shokuhin-Ippan-004) (M.H.) from the Ministry of Health, Labour and Welfare of Japan.

**Competing interests:** The authors have declared that no competing interests exist.

protein (PrP), designated as PrP$^{Sc}$, in the central nervous system is a distinct feature of prion diseases. Activation of microglia and astrocytes, which is often observed long before the clinical onset, is also a pathological hallmark of prion disease and is thought to be involved in its neuropathogenesis [1, 2]. In prion-infected mice, early synaptic dysfunction correlates with behavioral abnormalities, which precede histopathological alterations, including neuronal loss and the characteristic spongiform changes in neuropils [3, 4].

PrP$^{Sc}$ is a pathogenic and conformational isoform of cellular prion protein (PrP$^{C}$), which is encoded by the host gene *Prnp*. Since PrP-deficient mice are resistant to prion infection, PrP$^{C}$ is a key molecule in the pathogenesis of the prion diseases [5, 6]. Given the fact that PrP-deficient brain tissue is not damaged by exogenous PrP$^{Sc}$ produced in PrP-expressing tissue grafts nearby [7], and conditional knockout of the neuronal *Prnp* protects mice from prion disease [8, 9], the intraneuronal formation of PrP$^{Sc}$ seems requisite for prion pathogenesis. Therefore, it should be a top priority to identify the neuronal responses to prion propagation and to elucidate the mechanisms of neurodegeneration in prion diseases; as currently, a detailed mechanism is unclear. Mice inoculated with prions have been used for models of prion diseases as they reproduce the process of neurodegeneration that occurs during prion diseases in humans and animals. However, the presence of activated glial cells hampers the analysis of autonomous neuronal responses to prion propagation in neurons. Thus, neuronal cultures or *ex vivo* systems are required for finer analyses of the autonomous neuronal events that take place in response to PrP$^{Sc}$ formation. The use of immortalized neural cell lines are one option; however, only a few cell lines are capable of stable and consistent prion propagation [10, 11]. Unfortunately, these prion-susceptible cell lines show no sign of cytotoxicity as a consequence of persistent prion infection except for a single report using the GT1 hypothalamic cell line [12].

Primary neuronal cultures are thought to be useful *ex vivo* tools for the analysis of prion-induced neurodegeneration mechanisms [13–15]. Cronier et al. reported that primary cultures of cerebellar granule neurons (CGNs) from transgenic mice expressing ovine PrP with an astrocyte feeder layer have increased ratios of apoptosis when exposed to sheep scrapie agent [16]. In addition, Hannaoui et al. reported that decreases in microtubule-associated protein 2 (MAP2)-positive neuronal population occurred in murine CGNs, striatal, and cortical primary neurons after exposure to mouse-adapted scrapie prions. However, these primary neuronal cultures comprise not only neurons, but also glial fibrillary acidic protein (GFAP)-positive astrocytes that can account for up to 30% of the total cells [17]. This prompted us to use a neuron-enriched primary cerebral cortical culture in which the growth of astrocytes is minimized by anti-mitotic agent treatment. These primary cortical neurons are susceptible to mouse-adapted scrapie prions and produced PrP$^{Sc}$ similar to that in brains; neuronal PrP$^{Sc}$ localizes to cell surfaces and neurites, rather than intracellular vesicles, and also possesses its N-terminus [18]. In addition to cortical neurons, in the current study, we examined thalamic neurons which allow us to address possible differences in the vulnerability of neurons derived from different brain regions to prion propagation.

The involvement of unfolded protein response (UPR) has been suggested as one of the mechanisms for prion-induced neurodegeneration. UPR is an elaborate set of cellular responses to misfolded proteins overloading the endoplasmic reticulum (ER) [19], and is implicated in the pathophysiology of various protein misfolding diseases [20]. Of the three well-identified initial signal transducers of UPR, protein kinase RNA-activated-like ER kinase (PERK) is activated via autophosphorylation of a cytoplasmic domain when a luminal sensor domain detects ER stress. PERK then phosphorylates the α-subunit of eukaryotic initiation factor 2 (eIF2α), blocking its function in the 43S-translation-initiation complex, resulting in repression of global protein synthesis [21, 22]. Conversely, during ER stress, activating

transcription factor 4 (ATF4) is selectively translated and upregulates stress-responsive genes such as ATF3 (*Atf3*) [23], DNA damage-induced protein 34 kDa (GADD34/*Ppp1r15A*) [24], and the pro-apoptotic C/EBP homologous protein (CHOP/*Ddit3*) [24, 25]. While GADD34 plays a role in the negative feedback to dephosphorylate eIF2α, CHOP sensitizes cells that cannot keep up with persistent ER stress for cell death by down-regulating the anti-apoptotic protein Bcl-2, and by perturbation of cellular redox systems [26, 27]. With regard to the pathogenesis of prion diseases, upregulation of ER chaperones and enzymes such as Grp58, Grp78, and Grp94, which indicates the induction of UPR, has been reported in the brain of patients with sporadic and variant CJD, as well as prion-infected mice [28, 29]. Moreno et al. reported the involvement of PERK-eIF2α pathway during neurodegeneration in prion diseases; overexpression of GADD34, which reduced the phosphorylation levels of eIF2α, rescued mice from disease progress. Conversely, inhibition of eIF2α dephosphorylation exacerbated neurotoxicity in prion-infected mice [30, 31]. It is still an open question as to whether production of PrP$^{Sc}$ in neurons directly induces ER stress, and if it does, whether neuronal PrP$^{Sc}$ activates neurotoxic signaling pathways downstream of the PERK-eIF2α pathway.

In the present study, we analyzed the consequences of autonomous neuronal responses on prion propagation using neuron-enriched cortical and thalamic primary cultures. Although no apparent neuronal loss or neurite loss was associated with the continuous production of PrP$^{Sc}$ in the primary neurons, we found that phosphorylation of PERK was enhanced in prion-infected cortical neurons in accordance with the amount of PrP$^{Sc}$ in a cell. Interestingly, this did not fully activate the down stream pathway of UPR.

## Materials & methods

### Antibodies

For PrP$^{Sc}$-specific detection in immunofluorescence assay (IFA), mouse monoclonal antibodies (mAbs) 132 and 8D5 that recognize mouse PrP amino acids 119–127 [32] and 31–39 [33], were used. Anti-PrP mAb 31C6 that recognizes mouse PrP amino acids 143–149 [32] was used to detect proteinase K-resistant PrP (PrP-res) by immunoblotting. Other commercially available antibodies used for IFA and immunoblotting are listed in Table 1.

### Preparation of neuron-enriched primary cultures

All of the procedures for animal experiments were conducted in accordance with protocols approved by the Institutional Committee for Animal Experiments at Hokkaido University (approval No. 18–0110, approved until end March 2020). Hokkaido University provides the training for animal care and use twice a year. The animal care and use program at Faculty of Veterinary Medicine, Hokkaido University, has been accredited by AAALAC International since 2008. Primary cortical and thalamic neurons were prepared from ICR mouse embryos of embryonic day 14 (pregnant mice were purchased from Japan Clea Inc.) as described previously [18]. Pregnant mice were sacrificed under the anesthesia with Sevoflurane and embryos were recovered and cerebral cortices and thalami were collected from embryonic brains. Thirty pregnant ICR mice were used in this study. For neuron-enrichment, meninges were carefully removed and cells were treated with AraC (Sigma-Aldrich) at a final concentration of 0.25 μM from 4 to 7 days in vitro (div). At 7 div, microsomal fractions that were prepared from brains of terminally ill mice infected with mouse-adapted scrapie Chandler and Obihiro strains were inoculated into cultures at an amount equivalent to 5 ng of PrP-res per $10^5$ cells. Inoculum prepared from brains of age-matched healthy mice was used for mock-infection. At 4 days post infection (dpi), the media were completely replaced with fresh media without AraC. In some experiments, uninfected neurons were treated with tunicamycin (Sigma-

**Table 1. Commercially available antibodies used in this study.**

| Antigen | Host | poly/mono | Clone | Manufacturer | Product No. |
|---|---|---|---|---|---|
| ATF4 | Rabbit | mAb | D4B8 | CST | #11815 |
| Beta-III tubulin | rabbit | pAb | | abcam | ab18207 |
| Bip | Rabbit | pAb | | abcam | ab21685 |
| CHOP | Rabbit | mAb | D46F1 | CST | #5554 |
| eIF2-alpha | Rabbit | mAb | D7D3 | CST | #5324 |
| GluR1(AMPA subtype) | rabbit | pAb | | abcam | ab31232 |
| microtubule-associated protein 2 (MAP2) | Chicken | pAb | | abcam | ab5392 |
| NeuN | rabbit | mAb | EPR12763 | abcam | ab177487 |
| NMDAR1 | rabbit | mAb | EPR2481(2) | abcam | ab109182 |
| p-EIF2S1(S51) | Rabbit | mAb | E90 | abcam | ab32157 |
| PERK | Rabbit | mAb | C33E10 | CST[a] | #3192 |
| p-PERK(Thr980) | Rabbit | mAb | 16F8 | CST | #3179 |
| PSD95 | mouse | mAb | 7E3-1B8 | abcam | ab13552 |
| SAP102 | rabbit | pAb | | abcam | ab152132 |
| SNAP25 | rabbit | pAb | | abcam | ab5666 |
| synaptophysin | rabbit | mAb | EP1098Y | abcam | ab52636 |
| syntaxin1a | rabbit | pAb | | abcam | ab41453 |
| VAMP2 | rabbit | pAb | | abcam | ab3347 |

[a) Cell Signaling Technology

Aldrich) at a final concentration of 5 μg/ml for 24 hr just before the experiments to induce ER stress and UPR.

## Immunoblot analysis

Primary neurons were lysed with lysis buffer [34] and protein concentrations were measured with a DC protein assay (Bio-Rad). Cell lysates were subjected to SDS-PAGE on 8% or 15% polyacrylamide gels. For immunoblot analysis of UPR-related proteins, neurons were directly lysed in SDS-sample buffer containing a cOmplete protein inhibitor cocktail and a PhosSTOP phosphatase inhibitor cocktail (Roche). Proteins were transferred to polyvinylidene difluoride membranes and the blots were probed with appropriate antibodies followed by chemilumines-cence detection as previously described [34]. βIII-tubulin staining was used as an internal marker for normalization.

## Immunoblotting for PrP-res

Proteinase K digestions were carried out using 30 μg of total protein from each lysate with 20 ug of proteinase K for 10 min at 37˚C. SDS-PAGE, immunoblotting, and chemiluminescence detection were performed as previously described [34, 35]. PrP-res was detected by direct immunostaining with Fab fractions of mAb 31C6 that were genetically conjugated with human placental alkaline phosphatase [36].

**Immunofluorescence staining of primary neurons.** Immunostaining of primary neurons was performed as previously described [18, 37]. Briefly, neurons were fixed with 4% para-formaldehyde-PBS and permeabilized with 0.1% TritonX-100 in PBS. For PrP^Sc-specific staining with mAb 132, cells were treated with 5 M guanidine thiocyanate (GdnSCN) at RT for 10 minutes, while GdnSCN treatment was omitted for PrP^Sc-specific staining with mAb 8D5. After blocking with 5% fetal bovine serum-PBS, cells were incubated with primary antibodies

at 4˚C overnight. Appropriate second antibodies conjugated with Alexa Fluor dyes (Molecular Probes) were selected for each experiment, and nuclei were counterstained with 4',6-diami-dino-2-phenylindole (DAPI). A LSM700 inverted confocal microscope and ZEN2009 software (ZEISS) were used for image acquisition, except for the experiments in Fig 2 described in the next section.

## Image analysis of cell density and neurite outgrowth

Neurons cultured on ibidi μ-plates were immunostained for NeuN and MAP2 with DAPI counterstaining as described above. Images were acquired at 9 fixed positions in each well configured by XY-coordinates prior to experiment (in total $1.36 \times 10^6$ $\mu m^2$ from one well) using an Olympus IX71 fluorescence microscope with ×10 objective lens operated by MetaMorph software version 7.8.4.0 (Molecular Devices) using multiple stage positions and auto focus functions. MetaMorph was also used for image analyses. Living neuronal nuclei were identified as DAPI-positive objects whose size was $\geq 60$ $\mu m^2$ and average NeuN intensities that were above a threshold. We used the same threshold for both mock-infected and prion-infected preparations in each experiment. Neurite density was measured using a threshold function and was represented as the surface coverage (%) by MAP2-positive area. In these measurements, an average of the 9 positions was acquired to calculate a representative score of the well, and 9 wells from 3 independent experiments were used for each group.

## Quantitative reverse transcription polymerase chain reaction (qRT-PCR)

Neurons cultured on 12-well plates were directly dissolved into TRIzol reagent (Invitrogen) and total RNA was isolated according to the manufacturer's instructions. RNA was resuspended into nuclease-free water (Invitrogen), and the concentration was determined using a Nano Drop spectrometer (Thermo Scientific). cDNA was synthesized from 300–600 ng of total RNA using First-Strand cDNA Synthesis kit (GE Healthcare) with pd(N)$_6$ primer. Quantitative PCR was carried out using TaqMan Fast Universal PCR Mater Mix and TaqMan Probes (Applied Biosystems) on a 7900HT Fast Real-Time PCR system (Applied Biosystems). The TaqMan Probes used in this analysis are listed in Table 2. Data was analyzed by delta-delta cycle threshold method using *Tubb3* (βIII-tubulin) as an internal control.

## Preparation of recombinant adeno-associated viral vector (rAAV)

To visualize neurons in primary cultures, we used a rAAV expressing enhanced green fluorescence protein (EGFP) under the control of the human synapsin promoter. rAAV was prepared as described by Challis et al. [38] with slight modifications. Briefly, the viral genome was packaged into AAV-PHP.eB capsid by triple-transfection of HEK293T cells with three plasmids; pAAV-hSyn-EGFP was a gift from Bryan Roth (Addgene, plasmid #50465), pUC-mini-iCAP-PHP.eB was a gift from Viviana Gradinaru (Addgene, plasmid #103005) [39], and pHelper vector in AAVpro® CRISPR/SaCas9 Helper Free System (AAV2) (Takara Bio).

**Table 2. Commercially available TaqMan Probes used in this study.**

| Target gene | Product No. | Target gene | Product No. |
|---|---|---|---|
| Atf3 | Mm00476033_m1 | Ppp1r15A | Mm01205601_g1 |
| Ddit3 | Mm01135937_g1 | Snap25 | Mm01276449_m1 |
| Dlg4 | Mm00492193_m1 | Stx1a | Mm00444008_m1 |
| Gria1 | Mm00433753_m1 | Syp | Mm00436850_m1 |
| Hspa5 | Mm00517691_m1 | Tubb3 | Mm00727586_s1 |

rAAV-PHP.eB-hSyn-EGFP was purified by iodixanol-density gradient centrifugation using a the SW32 Ti rotor at 175,000g (32,000 rpm) for 6.5 hr at 18°C followed by ultrafiltration using Amicon Ultra-15 Ultracel-100K columns (Merck Millipore). Titration of rAAV was performed using quantitative PCR. To prepare the viral DNA sample, rAAV was digested with PK at 50 μg/ml at 50°C overnight. PK digestion was stopped by adding Pefabloc SC (Roche) at a final concentration of 2.2 mM and boiling for 20 min. Quantitative PCR was carried out using a custom-designed TaqMan Probe targeting the woodchuck hepatitis posttranscriptional regulatory element sequence with TaqMan Fast Universal PCR Mater Mix (Applied Biosystems). Linearized pAAV-hSyn-EGFP (5,265 base pairs) with *Pvu*I digestion was used as the DNA standard. Of note, 10 ng of the linearized dsDNA fragments correspond to $7.4 \times 10^9$ ssDNA molecules.

## Quantification of the signal frequency of phosphorylated PERK (p-PERK)

Z-series of confocal images (z-stack) from the bottom to the top of cell bodies of neurons immunostained for p-PERK, MAP2, and PrP$^{Sc}$ (with mAb 8D5) were acquired at 0.78 μm steps with a ×630 final magnification. The z-stacks were reconstructed by 3D rendering to build segmented objects designated as "surface" using the Surface function of Imaris software version 7.6.1. (Bitplane). MAP2-surface (cell bodies), DAPI-surface (nuclei), mAb 8D5-surface (PrP$^{Sc}$), and p-PERK-surface (p-PERK granules) were reconstructed. Neurites were manually pruned away to separate individual neurons. The number of p-PERK surfaces inside the cell body (MAP2 surface) but outside the nucleus (DAPI surface) of each neuron was counted and represented as the signal frequency (granule/cell). The number of PrP$^{Sc}$ surfaces inside or adherent to the cell body was also counted, which was used to classify prion-infected neurons into three subpopulations according to the amount of PrP$^{Sc}$ as described in Fig 5.

## Quantification of the density of synaptic terminals

Neurons transduced with the rAAV vector expressing EGFP were immunostained for PrP$^{Sc}$ (with mAb 8D5) and PSD95 or synaptophysin. For the double staining of PrP$^{Sc}$ with PSD95, the immunostaining procedure was slightly modified since both of the antibodies are mouse IgGs. First, immunostaining for PSD95 was completed by indirect method, then free Fab regions of the anti-mouse IgG secondary antibodies were blocked with a mouse IgG isotype control antibody to avoid cross-reaction with mAb 8D5. Alexa Fluor 555-conjugated mAb 8D5 was used for the PrP$^{Sc}$-specific immunostaining [18]. Z-stacks of EGFP-expressing neurons were acquired from the bottom to the top of neurites at 0.78 μm steps with a 63× objective lens and 0.5× digital zoom, which allowed us to obtain images with a 203 μm × 203 μm at ×315 final magnification. The surfaces of signals from EGFP (neurons), DAPI (nuclei), mAb 8D5 (PrP$^{Sc}$) and PSD95- or synaptophysin-positive granules (post- or pre-synaptic terminals), were reconstructed using Imaris software. The number of PSD95 or synaptophysin surfaces inside EGFP surfaces were counted and divided by the volume of each EGFP surface to represent the densities of post- and pre-synaptic terminals for each neuron (number of terminals/cell volume of 1000 voxels). The number of PrP$^{Sc}$ surfaces was also counted as described above.

## Results

### Prion propagation in primary cultures of mouse cerebral cortical and thalamic neurons

We previously reported that primarily cerebral cortical neuronal cultures (CxN) of ICR mouse embryos propagated mouse-adapted scrapie prions (22L, Chandler, and Obihiro strains) [18].

In the current study, we prepared primary neurons from thalamus (ThN) as well as the cortex by considering any differences in the efficiency of prion propagation *ex vivo*. To analyze autonomous neuronal events triggered by PrP$^{Sc}$ production in neurons, primary cultures were briefly treated with AraC to suppress glial cells growth (hereafter referred to as neuron-enrichment). Proportion of neuronal cells in primary cultures were shown in S1 Fig. NeuN-positive, GFAP-positive, and NeuN- and GFAP-double negative cells were 89.3%, 3.5%, and 7.2%, respectively in AraC treated, mock-infected CxN. GFAP-positive cells were 13.8% in CxN without AraC treatment, indicating successful reduction of astrocyte growth in our CxN. Similar tendency was observed in ThN. Immunoblotting showed linear increases in PrP-res from 7 days post infection (dpi) to 28 dpi, in CxNs and ThNs exposed to Chandler and Obihiro prions, demonstrating successful prion propagations. As expected, we did not detect PrP-res in mock-infected neurons (Fig 1A and 1B). PrP-res levels in CxNs were equivalent to those in ThNs for both prion strains tested at 21 dpi (Fig 1C). The level of PrP-res in Obhiro-infected neurons appeared to be higher than that in Chandler-infected neurons; however, this is probably due to difference in intrinsic PK-resistance of PrP$^{Sc}$ in each prion strain [40]. PrP$^{Sc}$-specific immunostaining with mAb 132 displayed PrP$^{Sc}$ clinging to neuronal cell bodies and neurites in prion-infected CxNs and ThNs (Fig 1D). The frequencies of PrP$^{Sc}$-positive neurons (Fig 1D, filled arrowheads) in Obihiro-infected CxN and ThN were 42.9% and 54.8%, respectively, when examined at 200× final magnification (21 dpi). Notably, some PrP$^{Sc}$-positive neurons possessed a swarm of filamentous PrP$^{Sc}$ staining around their cell bodies and neurites (fPrP$^{Sc}$ neurons, Fig 1D). fPrP$^{Sc}$ neurons accounted for 10.0% and 7.5% of the total neurons in Obihiro-infected CxN and ThN (21 dpi), respectively.

## Influence of prion propagation on the densities of neuronal cells and neurites

CxN and ThN successfully propagated Chandler and Obihiro prions (Fig 1). Thus, we examined whether production of PrP$^{Sc}$ could cause any immediate alteration in CxN and ThN. Microtubule-associated protein 2, MAP2, is mainly distributed over the dendrites and cell bodies of mature neurons and is widely used as a neuronal marker. NeuN is an alternative splicing factor of pre-mRNAs named *Rbfox3*, and is also a reliable marker for mature neurons. Since NeuN is detected in neuronal nuclei and disappears when cells die, immunostaining for NeuN is a convenient way for counting live neurons in primary culture where dead cell populations are inevitable over time. We immunostained for MAP2 and NeuN, followed by quantitative image analysis to evaluate the number of neurons and their density of neurites in CxN and ThN. As shown in S1 Fig, there were time-dependent changes in the number of NeuN-positive cells and MAP2-positive neurite extensions; CxN grew slowly and displayed the highest densities of neuronal cells at 14 dpi followed by continued elaboration into a dense neurite network, whereas ThN had the highest densities of both neuronal cells and neurites at 7 dpi. Taking into consideration experimental variations that exist between independently prepared primary neuronal cultures, measurements was normalized to the average of mock-treated neurons in the same experiment at each time point. Neuronal cell density was lower in prion-infected CxNs than in mock-infected CxN at 7 dpi, however, the decreasing trends were not continuous (Fig 2A); in fact, the density of neuronal cells in prion-infected CxNs were comparable to mock-infected CxN at 21 dpi when 42.9% of cells were positive for PrP$^{Sc}$ in Obihiro-infected CxN. Neurite densities of prion-infected CxNs were slightly lower than those of mock-infected CxN at all experimental timepoints, but neurodegenerative alterations, such as exacerbation of neurite sparseness, were not observed despite the rising levels of PrP-res (Fig 2A). In prion-infected ThNs, the number of neurons decreased to approximately 70% of

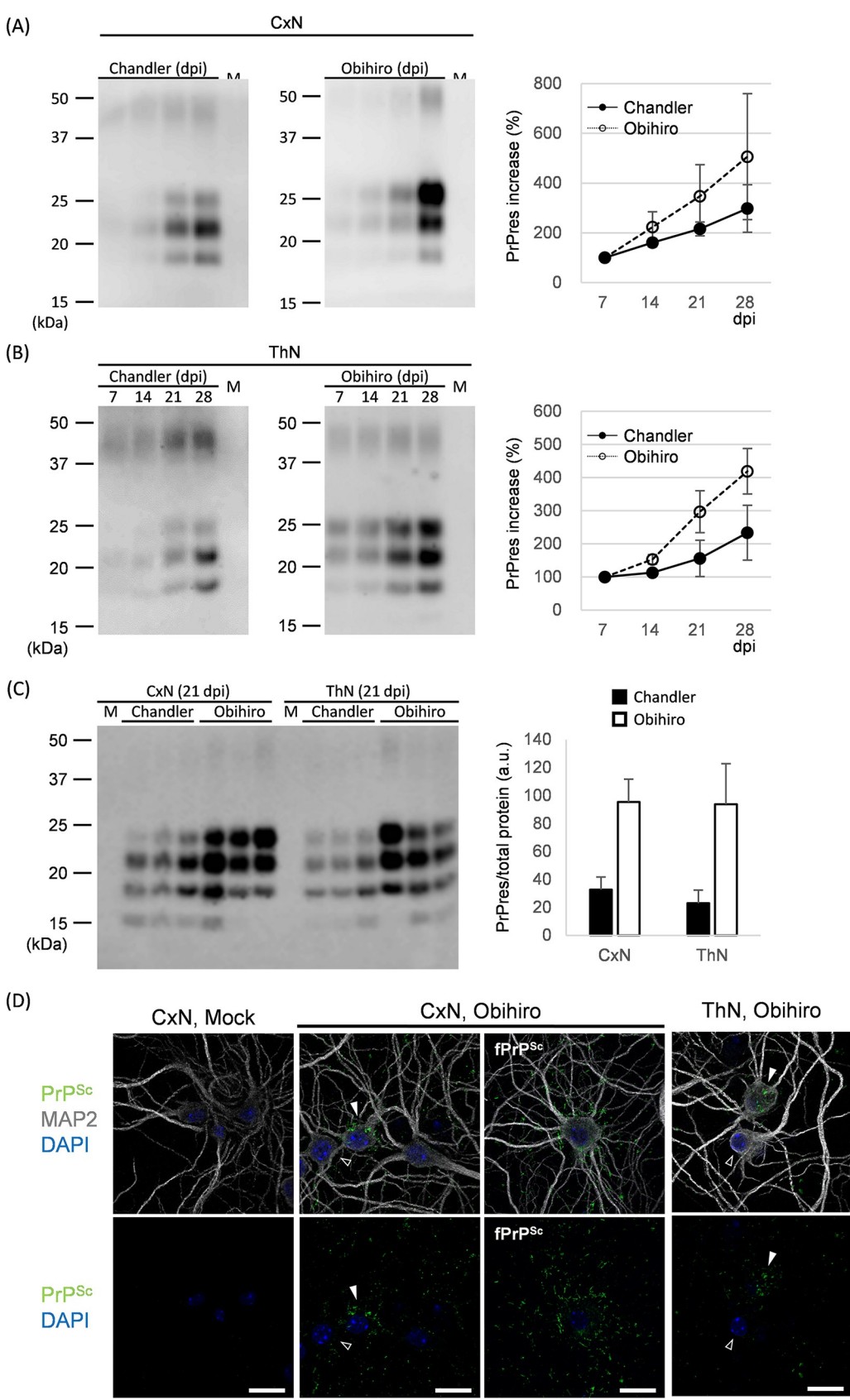

**Fig 1. Prion propagation in primary cultures of mouse cerebral cortical and thalamic neurons.** (A, B) Representative immunoblots for PK-resistant PrP$^{Sc}$ (PrP-res) in (A) cerebral cortical neurons (CxN) and (B) thalamic neurons (ThN), infected with mouse-adapted scrapie Chandler and Obihiro strains. Graphs show the results of quantification as a mean ± SE of three independent experiments performed in triplicate. Mean intensities at 7 dpi were used as benchmarks (100%) for time-course comparisons in order to show rate of prion propagation (detected as PrP-res) after the establishment of prion infection in the primary neurons: dpi, days post infection; M, mock-infected neurons at 28 dpi. (C) Comparison of PrP-res levels between neuronal types or prion strains at 21 dpi. The graph shows the mean ± SD for experiments performed in triplicate. (D) Immunofluorescent detection of PrP$^{Sc}$ in primary neurons at 21 dpi. Cells were denatured with guanidine thiocyanate but not digested with PK for PrP$^{Sc}$-specific detection by mAb132. Panels labeled fPrP$^{Sc}$ show neurons with a swarm of typical filamentous PrP$^{Sc}$ staining. Filled arrowheads, neurons with PrP$^{Sc}$ staining (PrP$^{Sc}$-positive neurons); open arrowheads, neurons with apparently no PrP$^{Sc}$ staining. Figures are shown as maximum intensity projection images created from z-stack confocal sections at ×315 final magnification. Bars, 20 μm.

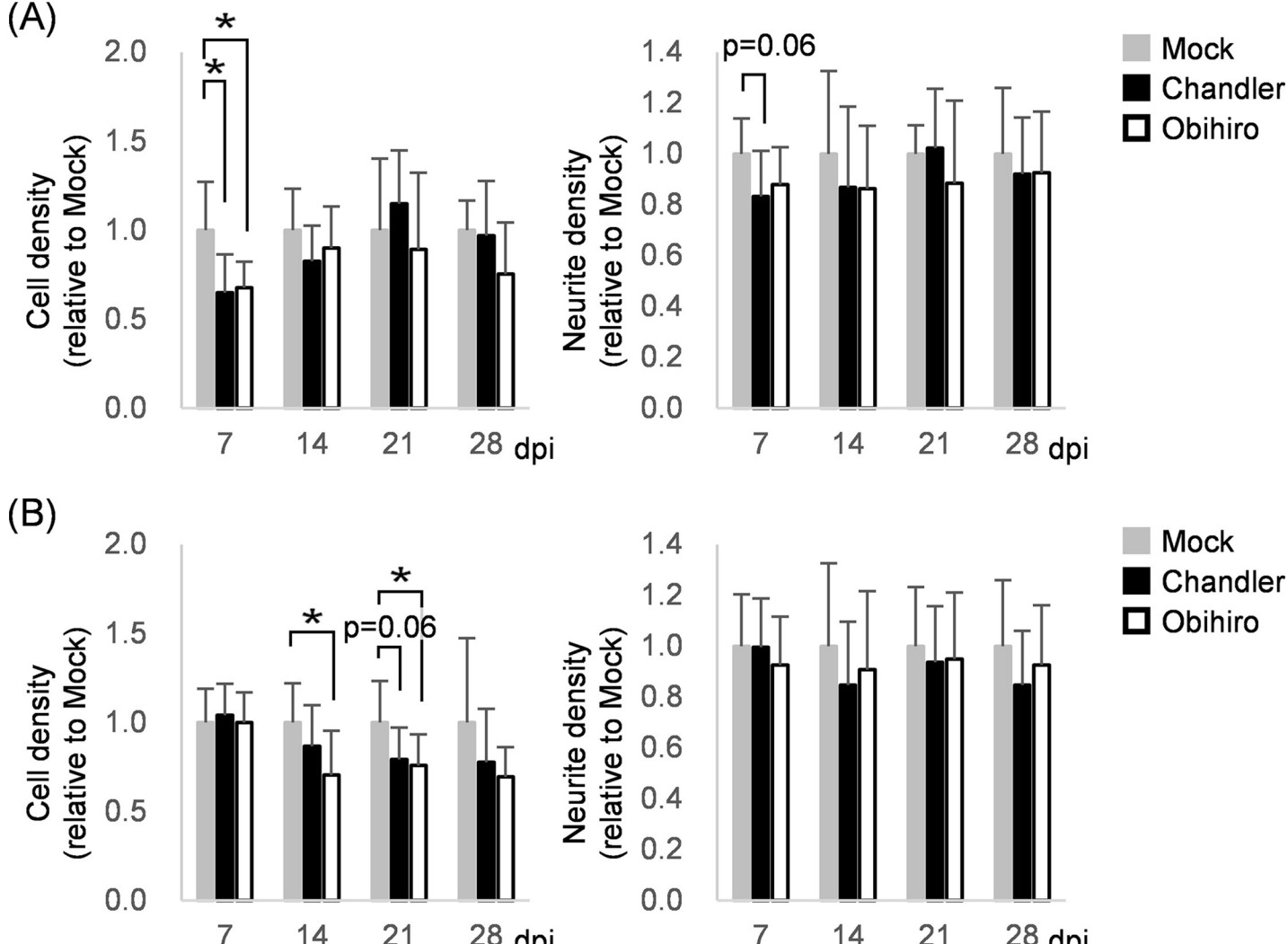

**Fig 2. Influence of prion propagation on the density of neuronal cells and neurites.** Neuronal cell density and neurite density of CxNs (A) and ThNs (B) were measured by immunofluorescence assay followed by image analysis using MetaMorph software. Cell density and neurite density were defined as the number of NeuN-positive nuclei per 0.01 mm$^2$ and the surface coverage (%) by MAP2-positive neurites, respectively, which are represented as relative values to the averages of corresponding mock-infected neurons at each dpi (mean ± SD of nine replicates from three independent experiments). *, $p < 0.05$ by Dunnett's test using mock-infected neurons as control groups.

mock-infected ThN by 14 dpi, implying the existence of subpopulation(s) that are more susceptible to acute prion toxicity [15, 41] *ex vivo* (Fig 2B). However, prion-infected ThNs did not show progressive neuronal loss despite continuous increases in PrP-res levels. Moreover, prion-infected ThNs maintained dense neurite-networks comparable to mock-infected ThNs throughout the experiment (Fig 2B), suggesting that the production of PrP$^{Sc}$ did not solely induce neuronal death nor neurite loss detectable by immunostaining for MAP2 and NeuN.

## ER stress and UPR in prion-infected neurons

The PERK-eIF2α pathway of UPR copes with disrupted proteostasis by restraining global protein synthesis via phosphorylation of PERK and eIF2α; reducing ER tasks and enabling cells time to refold or dispose of inaccurately synthesized proteins [19]. ATF4 is a key molecule in this pathway as it induces effector molecule production, including the pro-apoptotic protein CHOP [42]. The involvement of ER stress and the PERK-eIF2α pathway in prion pathogenesis has been demonstrated in prion-infected mice [30, 31], though it is still unclear whether prion propagation directly evokes ER stress and induces UPR in neurons. To examine the activation state of UPR during the continuous production of PrP$^{Sc}$ in CxN and ThN, we first analyzed phosphorylation levels of PERK and eIF2α, and the induction of ATF4 in Obihiro-infected neurons. Immunoblotting revealed that enhanced phosphorylation of PERK in prion-infected CxNs at 7 dpi and thereafter. The prion-infected CxNs also had a tendency to increase the phosphorylation level of eIF2α, although it was not statistically significant (Fig 3A and 3C). On the other hand, phosphorylation levels of PERK and eIF2α in Obihiro-infected ThN were comparable to those of uninfected ThN (Fig 3B and 3C), despite the comparable amounts of PrP-res in Obihiro-infected ThN and CxN (Fig 1A–1C). In contrast to the enhanced phosphorylation of PERK, protein levels of ATF4 in prion-infected CxN and ThN were not elevated (Fig 3A–3C). We further examined gene expression levels for *Atf3*, *Ddit3*, *Ppp1r15A*, and *Hspa5*, which encode ATF3, CHOP, GADD34, and Bip, respectively, all of which are known to be up-regulated during ER stress through the PERK-ATF4 axis [23, 24, 42–44]. Upregulation of these genes was not observed in prion-infected CxNs and ThNs, except for Hspa5 in prion-infected CxNs at 21dpi (Fig 3D), which can be upregulated by other pathways of UPR, such as ATF6-mediated signaling [45]. As a whole, the results of transcriptional analysis were consistent with no increase ATF4 protein levels. Ratios of p-PERK to total PERK and p-eIF2α to eIF2α were calculated (S3 Fig). Although differences were not statistically significant, p-PERK/total PERK ratios tended to be higher in prion-infected CxN and ThN than mock-infected CxN and ThN. Ratios of p-eIF2α/ total eIF2α did not differ between prion-infected and mock-infected primary neurons. Taken together, the results suggest that phosphorylation of PERK was induced in response to prion propagation in CxNs, which may reflect ER stress in prion-infected CxNs. However, the downstream pathway of PERK was not activated.

## Influence of prion propagation on the amounts of post- and pre-synaptic proteins

Synaptic loss and disfunctions have been observed in the brain of prion-affected animals prior to neuronal loss [4, 46]. Of those alterations, decreases in the protein levels of several synaptic proteins such as SNAP25 and PSD95 have been reported in association with sustained activation of the UPR PERK-eIF2α pathway [30]. To confirm whether prion propagation has any influence on synaptic proteins in cultured neurons, we analyzed a total of 8 post- and pre-synaptic proteins by immunoblotting (Fig 4): post-synaptic AMPAR and NMDAR1 (glutamate receptors), and PSD95 and SAP102 (scaffold proteins); pre-synaptic syntaxin1a, SNAP25, and VAMP2 (SNARE proteins involved in intracellular membrane fusion during exocytosis of

neurotransmitters), and synaptophysin (an integral membrane protein of synapse vesicles). At 7 dpi, although most differences were not statistically significant except for SNAP25 ($p < 0.05$), the levels of these synaptic proteins in Chandler- and Obihiro-infected CxNs

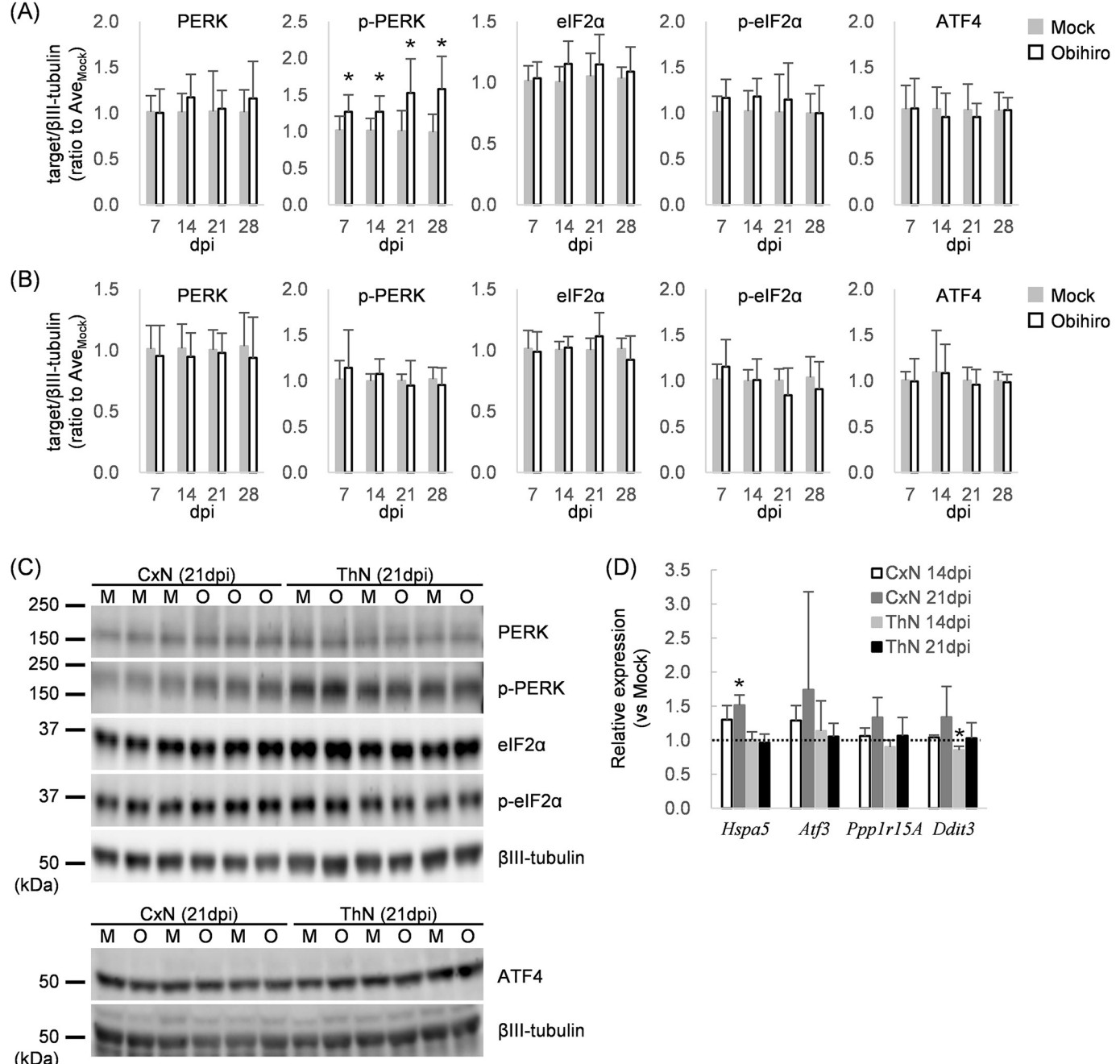

**Fig 3. ER stress and unfolded protein responses in prion-infected neurons.** (A, B) Immunoblot analysis of molecules involved in the PERK-eIF2α pathway of CxNs (A) and ThNs (B). Protein levels are represented as relative values to the average of mock-infected neurons at each dpi (mean ± SD of 6 replicates from 2 independent experiments; p-PERK, phosphorylated PERK; p-eIF2α, phosphorylated eIF2α; *, $p < 0.05$ by Student's $t$ test). (C) Representative blots used for quantification shown in (A) and (B). βIII-tubulin from the same blots were used as internal controls: M, mock-infected neurons; O, Obihiro-infected neurons. For the detection of βIII-tubulin in the analysis of ATF4, peroxidase conjugated anti-rabbit IgG antibodies used for the detection of ATF4 was deactivated with 15% hydrogen peroxide solution, then βIII-tubulin was detected with peroxidase-conjugated anti-chicken IgY second antibody. (D) Expression of genes in the PERK-ATF4 axis analyzed by qRT-PCR. Relative expression levels in Obihiro-infected CxN and ThN to dpi-matched mock-infected neurons are shown as mean ± SD of 3 independent experiments. The broken line indicates relative expression levels = 1 (*, $p < 0.05$ by One sample $t$ test).

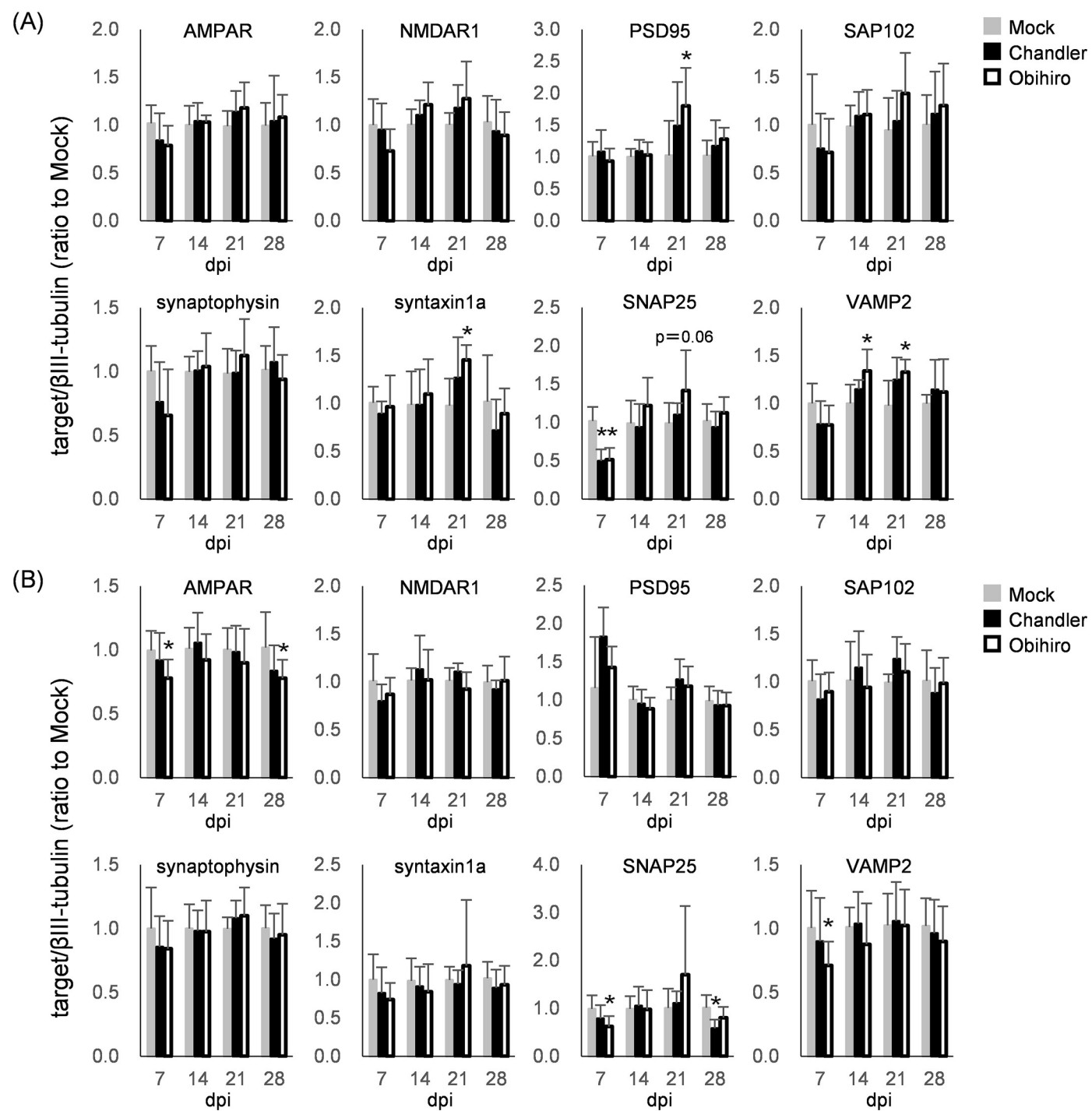

**Fig 4. Influence of prion propagation on the levels of post- and pre-synaptic proteins.** (A, B) Immunoblot analysis of post- and pre-synaptic proteins in CxNs (A) and ThNs (B). βIII-tubulin from the same blot was used as internal controls. Protein levels of post-(AMPAR, NMDAR1, PSD95 and SAP102) and pre-(synaptophysin, syntaxin1a, SNAP25 and VAMP2) synapse-associated molecules are represented as relative values to the average of mock-infected neurons at each dpi (mean ± SD of 7–8 biological replicates from 2 independent experiments; *, $p < 0.05$ by Dunnett's test using mock-infected neurons as control groups).

showed tendencies toward decreased levels in comparison to mock-infected CxNs (Fig 4A), which allowed us to speculate that prion propagation had a repressive influence on the expression of synaptic proteins in the early phase of prion infection. Intriguingly, this trend

unexpectedly reversed; leading to increased synaptic markers by 21 dpi. In particular, the levels of PSD95, syntaxin1a, and VAMP2 in Obihiro-infected CxNs were significantly higher than mock-infected CxNs. Of note, several studies conducted in the last decade using animal models and brain slices also reported that increased numbers of synaptic terminals or spines and the size of post-synaptic densities in the early phases of disease, either just before or after the first detection of PrP-res [30, 47, 48], which were similar to the autonomous neuronal events we observed.

Obihiro-infected ThNs at 7 dpi also had a tendency for decreased synaptic protein levels, similar to CxNs. The levels of AMPAR, VAMP2, and SNAP25 were significantly decreased ($p < 0.05$) compared to mock-infected ThNs (Fig 4B). The decrease in protein levels appeared to be restored by 21 dpi; however, transient increases in synaptic protein levels were less obvious in prion-infected ThNs when compared to CxNs. Taken as a whole, the results show a tendency for the loss of both post- and pre-synaptic proteins in the early stages of prion infection in primary neurons (at 7 dpi). Unexpectedly, the trend was not observed thereafter, despite the continuous prion propagation.

## Phosphorylation of PERK in PrP$^{Sc}$-positive neurons

Prion-infected CxNs and ThNs did not show any evident neuronal death or neurodegeneration in the current study. However, considering that the ratio of PrP$^{Sc}$-positive neurons as examined by IFA were 42.9% in CxNs and 54.8% in ThNs including fPrP$^{Sc}$ neurons (10.0% and 7.5%, respectively) at 21 dpi, we analyzed individual PrP$^{Sc}$-positive neurons to examine the effects of PrP$^{Sc}$ production on neurons more carefully. The mAb 8D5 has the unique property that it selectively binds to native PrP$^{Sc}$ in IFA without treating cells with chaotropic agents such as GdnSCN [18, 33], which enabled us to carry out a PrP$^{Sc}$-positive neuron-selective analysis.

We examined neurons in mock- and Obihiro-infected CxNs at 21 dpi for phosphorylated PERK (p-PERK) and PrP$^{Sc}$ by immunofluorescence staining to ascertain whether initial UPR events were triggered by PrP$^{Sc}$ production (Fig 5). Specificity of anti-phospho-PERK antibody in IFA was confirmed using tunicamycin-treated Neuro2a cells; fluorescent granular signals stained by this antibody appeared more in tunicamycin-treated Neuro2a cells than in untreated Neuro2a cells (S4 Fig). Granular stains of p-PERK were observed in the soma of mock-infected and Obihiro-infected neurons in CxNs at 21dpi, indicating that phosphorylation of PERK is not an event specific to prion infection (Fig 5A). Interestingly, however, intense granular stains of p-PERK were often observed in neurons with a larger number of granular and/or filamentous PrP$^{Sc}$ signals (Fig 5A, arrow). Quantitative image analysis of p-PERK and PrP$^{Sc}$ signals in each neuron identified a positive correlation between these two molecules in neurons in Obihiro-infected CxN (Fig 5B, correlation coefficient = 0.49). When neurons in Obihiro-infected CxNs were divided into three subpopulations according to the frequency of PrP$^{Sc}$ signal at its soma (PrP$^{Sc}$_low, PrP$^{Sc}$+, and PrP$^{Sc}$++), the frequency of p-PERK signals was significantly increased in PrP$^{Sc}$+ neurons, and PrP$^{Sc}$++ neurons when compared to either mock-infected neurons or PrP$^{Sc}$_low neurons (Fig 5C). These data strongly suggest that the production of PrP$^{Sc}$ caused ER stress to neurons and accelerated the phosphorylation of PERK, which is known to be the initial event for activating the PERK-ATF4 axis of UPR. The acceleration of PERK phosphorylation was also observed in PrP$^{Sc}$++ neurons of ThN (S5 Fig). We reexamined the expression of ATF4 by co-immunofluorescence staining to see if the PERK-ATF4 axis was activated in those neurons. While induction and nuclear localization of ATF4 were observed in both CxNs treated with tunicamycin, an ER stressor, no ATF4-positive neurons were observed in Obihiro-infected CxNs (Fig 5D).

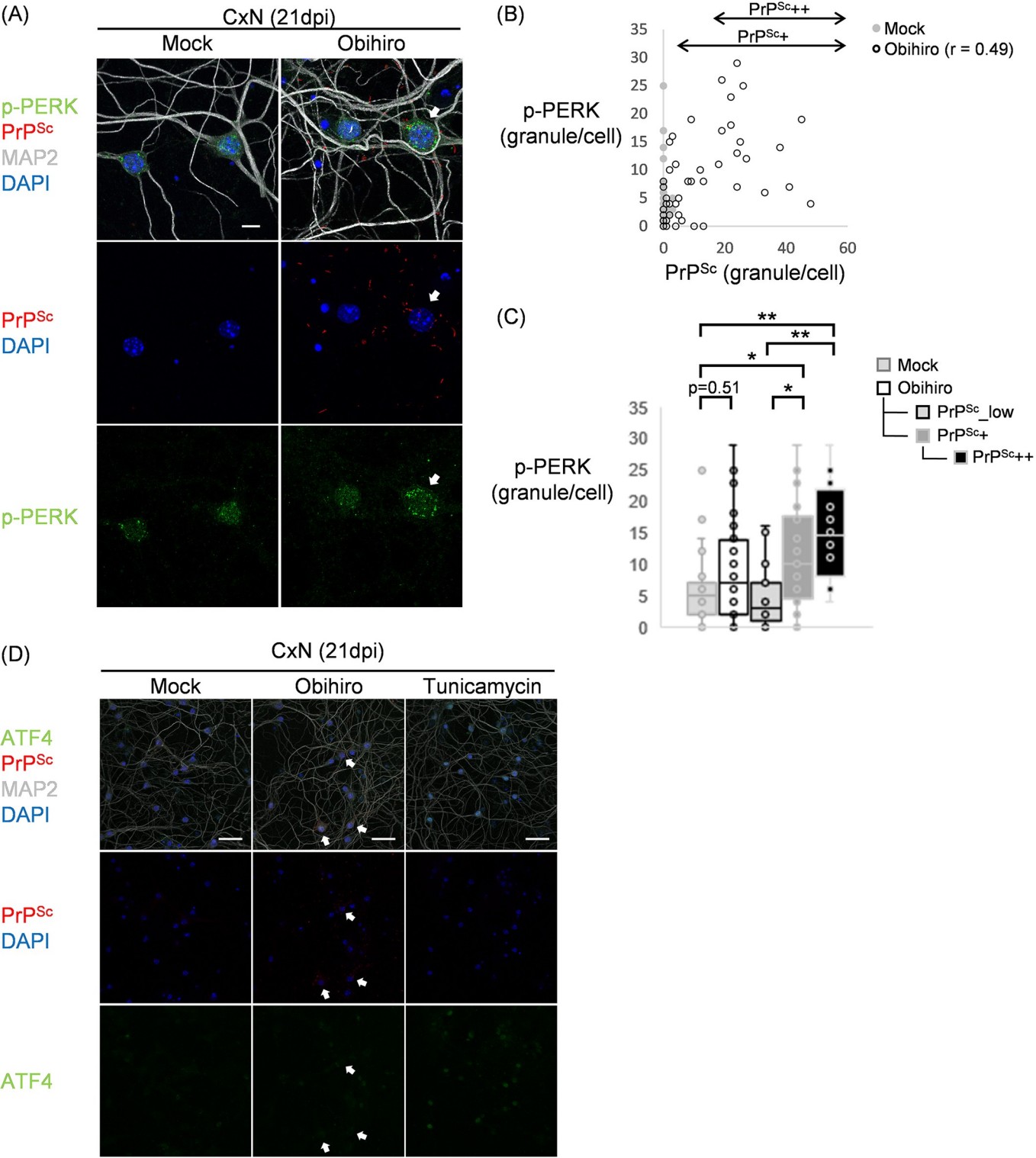

**Fig 5. Phosphorylation of PERK in PrP$^{Sc}$-positive neurons.** (A) Multiple immunofluorescence staining for p-PERK and PrP$^{Sc}$ with counter staining for MAP2 and nuclei (DAPI) in CxNs at 21 dpi. PrP$^{Sc}$ was stained with mAb 8D5. Figures are shown as maximum intensity projection images created from z-series stacks of confocal images at ×630 final magnification. Arrows, PrP$^{Sc}$++ neurons. Bar, 10 μm. (B, C) Quantification of p-PERK and PrP$^{Sc}$ in individual neurons by 3D-image analysis using Imaris software. Each dot in the scatter diagram (B) represents an individual neuron (r, correlation coefficient). In (C), Obihiro-infected CxNs were analyzed as a whole (Obihiro) and as three subpopulations classified by the frequency of PrP$^{Sc}$ signals at the soma. The definition of the three subpopulations and the number of cells used for this analysis is as follows: PrP$^{Sc}$_low (< 4 PrP$^{Sc}$ signals/cell, n = 19), PrP$^{Sc}$+ (≥ 4 PrP$^{Sc}$ signals/cell, n = 29,

including PrP$^{Sc}$++), and PrP$^{Sc}$++ ($\geq$ 18 PrP$^{Sc}$ signals/cell, n = 16). The number of cells in mock-infected CxNs and Obihiro-infected CxNs used for the analysis are n = 35 and n = 48 (the total of PrP$^{Sc}$_low and PrP$^{Sc}$+ subpopulations), respectively. Differences between the groups were analyzed by Steel-Dwass's multiple comparison tests (*, $p < 0.05$; **, $p < 0.01$). (D) Co-immunofluorescence staining for ATF4 and PrP$^{Sc}$ with counter staining for MAP2 and nuclei (DAPI) in CxNs at 21 dpi. PrP$^{Sc}$ was stained with mAb8D5. Uninfected neurons treated with tunicamycin were used as a positive control for ATF4-staining. The green fluorescence in mock- and Obihiro-infected CxNs is the autofluorescence from the condensed nuclei of dead cells. Arrows, PrP$^{Sc}$++ neurons. Bar, 50 μm.

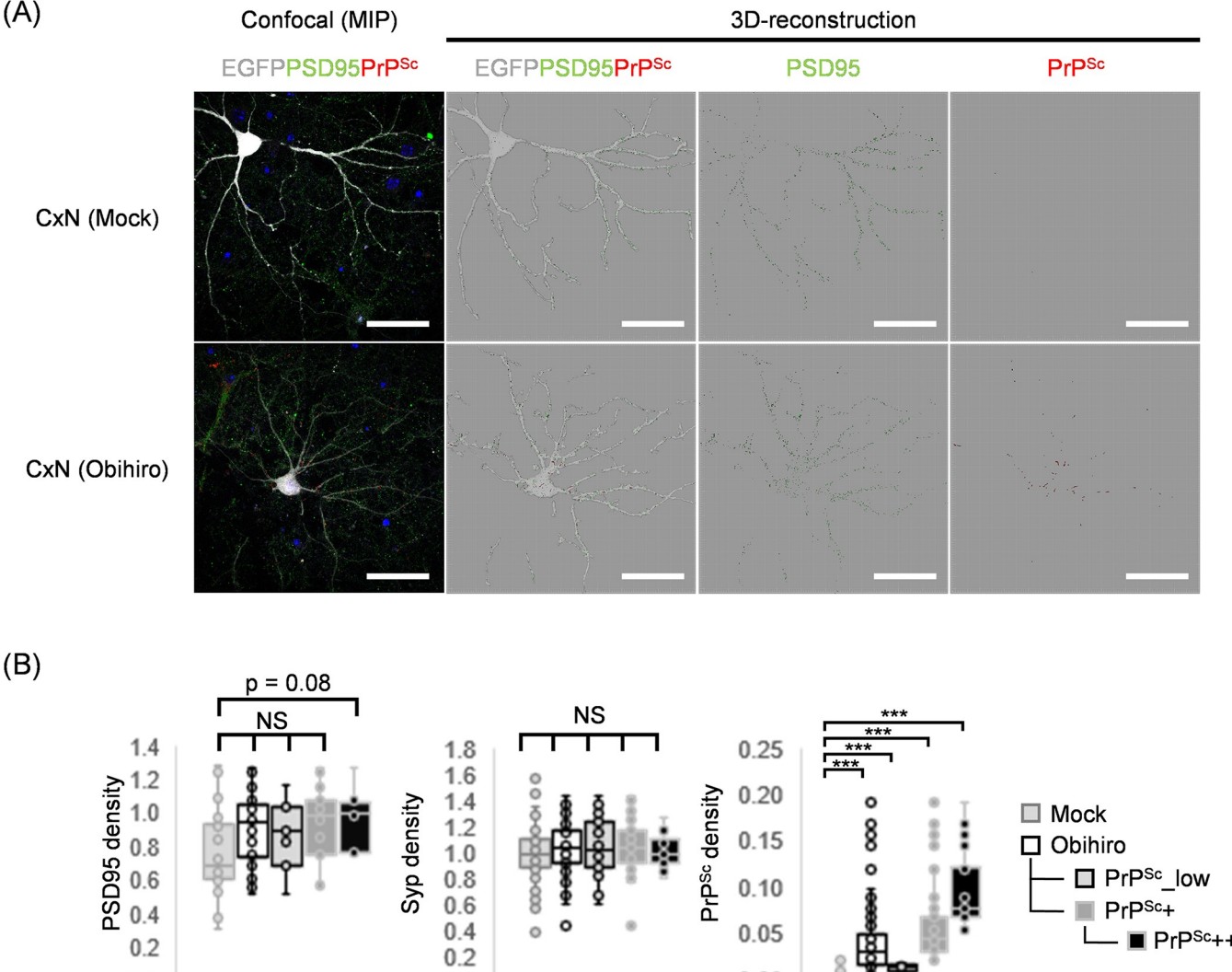

**Fig 6. Synapse densities of PrP$^{Sc}$-positive neurons.** Neurons were transduced with the adeno-associated viral vector for EGFP expression before immunostaining of PrP$^{Sc}$ and PSD95 or synaptophysin at 21 dpi. Z-stacks of confocal images at a final magnification of 315× were reconstructed in 3D-images for analysis using Imaris software. (A) Representative confocal images (MIP, maximum intensity projection) and reconstructed 3D-images (surfaces) of EGFP-expressing neurons immunostained for PrP$^{Sc}$ (with mAB 8D5) and PSD95 in CxNs. (B) Densities of PSD95-positive post-synaptic terminals, synaptophysin-positive pre-synaptic terminals, and PrP$^{Sc}$ in CxNs. Densities were defined as the number of signals per unit volume of cell. Obihiro-infected CxNs were analyzed as a whole (Obihiro) and as three subpopulations classified by the density of PrP$^{Sc}$ signals in soma and neurites. The rightmost graph shows the density of PrP$^{Sc}$ from a representative data set used for pre-synapse quantification. The numbers of neurons used for the measurements were as follows: for post-synaptic density measurements, 20 neurons from mock-infected CxNs and 26 neurons from Obihiro-infected CxNs were used (PrP$^{Sc}$_low, n = 7; PrP$^{Sc}$+, n = 19 including PrP$^{Sc}$++, n = 8). For pre-synaptic density measurements, 73 neurons from mock-infected CxNs and 78 neurons from Obihiro-infected CxNs were used (PrP$^{Sc}$_low, n = 26; PrP$^{Sc}$+, n = 52 including PrP$^{Sc}$++, n = 19). Differences between the groups were analyzed by a Steel's test using mock-infected neurons as control groups (NS, no significant differences; ***, $p < 0.001$).

## Synapse densities of PrP^Sc-positive neurons

In our last series of experiments, we examined synaptic alterations in PrP$^{Sc}$-positive neurons by co-immunostaining. The dense networks of neurites in the primary cultures made it difficult to identify processes from a single neuron by immunofluorescence staining for MAP2; therefore, EGFP was expressed in neurons using an adeno-associated viral vector. In this analysis we focused on the densities at post- and pre-synaptic terminals, i.e., the number of synaptic terminals in a unit volume of a cell, inside and in close proximity to EGFP-expressing neurons by 3D-measurement using a z-series of confocal images (Fig 6A). Obihiro-infected CxNs were divided into three subpopulations according to the density of PrP$^{Sc}$ signals as described above (Fig 6B, the right graphs). Quantitative image analyses of post- and pre-synaptic terminals, using PSD95 and synaptophysin as markers respectively, did not show any synaptic losses in Obihiro-infected CxNs at 21 dpi, even in PrP$^{Sc}$++ subpopulations (Fig 6B). The average density of PSD95-positive post-synaptic terminals was slightly higher in PrP$^{Sc}$++ neurons when compared to mock-infected neurons ($p = 0.08$); which may reflect the increased levels of PSD95 in Obihiro-infected CxN at 21 dpi detected by immunoblot analysis (Fig 4A).

## Discussion

Conversion of neuronal PrP$^C$ into PrP$^{Sc}$ is a central event that leads to neurodegenerative consequences in prion diseases [7–9]; however, the precise process of neurodegeneration and its molecular mechanisms remain unclear. Although the involvement of microglia in the neuropathogenesis of prion diseases is evident [49–51], little is known about autonomous neuronal responses to prion propagation, which may cause neuronal death. Hannaoui et al. reported that primary neuronal cultures from mouse cerebellum and cortex showed neuronal losses after prion exposure [17], suggesting that primary neuronal cultures are a versatile *ex vivo* model for analyzing the molecular mechanisms of neurodegeneration by prion infection. However, it is difficult to evaluate autonomous neuronal response to prion propagation due to the presence of astrocytes in those cultures, particularly if cultures are kept for long periods [17]. Thus, in the current study, we used neuron-enriched primary cultures from cortices and thalami by inhibiting astrocyte growth with AraC treatment, which inhibits astrocyte growth [18].

Previously, Cronier et al. observed an increase of apoptotic cells with fragmented nuclei in prion-infected CGNs cultured on an astrocyte feeder layer [16]. Hannaoui et al. reported decreases in the proportion of MAP2-positive neurons in prion-exposed CGNs from 14–21 days post exposure (dpe), as well as in striatal and cortical neuronal cultures from 7–11 dpe; GFAP-positive astrocytes accounted for 20–30% of cells when neuronal losses were observed [17]. In contrast to these studies, no continuous decline of NeuN-positive neurons or MAP-2-positive neurite densities in prion-infected cells were observed in CxNs or ThNs infected with either the Chandler or Obihiro prion strains (Fig 2A), in spite of the continuous increasing of PrP-res levels over 28 dpi. Differences in experimental conditions, including mouse strains used for neuronal cultures and prion strains, may account for some of the inconsistency of observed neurotoxicity. One of critical differences between Hannaoui's study and the current study was presence of astrocytes. Astrocytes can be infected with prions and support the propagation and transmission of prions in cell cultures [16, 52–54] and change the microenvironment, which accelerates degeneration of prion-infected neurons [55]. Thus, it is possible that efficient prion propagation in primary neuronal cultures with astrocytes attributes to the differences in observed neurotoxicity.

Induction of ER stress, which is recognized by upregulation of ER chaperon proteins, has been reported in the brains of patients with sporadic or variant CJD [28, 29] and prion-

infected mice [28]. The involvement of UPR, in particular the PERK-eIF2α-ATF4 pathway, in the pathogenesis of prion diseases has been demonstrated by the inhibition of the PERK activity and/or the reduction of eIF2α phosphorylation levels which successfully prolonged the survival of prion-infected transgenic mice overexpressing PrP [30, 31]. Thus, we analyzed molecules involved in the PERK-eIF2α-ATF4 pathway of UPR to assess whether UPR was evoked in prion-infected CxNs and ThNs as an autonomous neuronal response. Although no clear tendency of activation of this pathway was observed when cultures were analyzed as a whole (Fig 3), a finer analysis of individual neurons by IFA identified a positive correlation between the amounts of PrP$^{Sc}$ and p-PERK in cortical neurons at 21 dpi, indicating that the production of PrP$^{Sc}$ directly provoked neuronal ER stress and enhanced the phosphorylation of PERK as an autonomous neuronal response (Fig 5A–5C). ATF4 is a pivotal transcription factor for full activation of the PERK-eIF2α pathway ultimately transducing ER stress into a CHOP-mediated proapoptotic phase [27]. Indeed, time-dependent increases of CHOP levels and increased translation of ATF4 in the brain of prion-infected transgenic mice suggests the activation of this axis with the disease progression [30]. Interestingly, neither induction of ATF4 nor upregulation of its downstream genes was observed, even in PrP$^{Sc}$++ neurons in prion-infected CxN (Fig 5C). Our data suggests that ER stress and PERK phosphorylation are enhanced by prion propagation as an autonomous neuronal response, whereas activation of downstream cascades in the PERK-eIF2α-ATF4 pathway will not take place as a consequence of autonomous neuronal response to prion propagation.

Given that prion diseases have a long preclinical phase of prion propagation before exhibiting clinical manifestations *in vivo* [56], we cannot exclude the possibility that level or duration of PrP$^{Sc}$ accumulation in our prion-infected neurons is enough to trigger ER stress but not sufficient to activate UPR that leads to neurodegeneration. It is probable that activated microglia and/or astrocytes play intrinsic roles in the process of neurodegeneration in prion diseases [57, 58]. Depletion and inhibition of microglial activity alters the disease progression of prion-infected mice [49, 50, 59]. Astrogliosis may also modify the neuropathological processes of prion diseases [60, 61]. Recently, Smith et al. reported that changes in astrocyte secretome by UPR-response in astrocytes as the cause of non-neuron autonomous neurodegeneration in prion diseases, although changes in astrocytes by prion-induced UPR, which may directly cause the neurodegeneration, are not fully understood [62]. There are several pieces of evidence for so-called "cross talk" [63, 64] between neurons and glial cells that regulates their activation states; transducing both neuroprotective and neurotoxic signals under pathological conditions. Therefore, additional stimuli, possibly from activated microglia or astrocytes, may be required for the full activation of the PERK-eIF2α-ATF4 pathway in the brains of prion-infected mice to produce the neurodegeneration observed in prion diseases.

Decrease in levels of synaptic proteins has been observed in brains of prion-infected mice [4, 46, 65]. It is known that phosphorylation of eIF2α by p-PERK induces the global repression of translation as a consequence of UPR [21], which targets include several synaptic proteins [30]. Thus, we examined the expression of post- and pre-synaptic proteins in individual neurons at 21 dpi; however, no differences were observed in the densities of PSD95- or synaptophysin-positive granular signals between mock-infected and prion-infected neurons even in PrP$^{Sc}$++ neurons (Fig 6). These results suggest that not only selective translation of ATF4, but also repression of global protein synthesis was not autonomously induced in prion-infected neurons by PERK phosphorylation, consistent with no differences in p-eIF2α levels being observed between mock- and prion-infected neurons at 21 dpi (Fig 3). Incidentally, at 7 dpi, we found that eIF2α phosphorylation levels were significantly higher, which coincided with the decreased expression of several synaptic proteins in prion-infected CxNs versus mock-infected CxNs in immunoblot analyses. This suggests that activation of UPR via p-PERK is

somehow limited to a relatively early phase after exposure to prions, even though the ER stress and phosphorylation of PERK are sustained in prion-infected neurons. The importance of early PERK-eIF2α pathway activation in neurodegenerative disorders has been also indicated by p-PERK in pre-tangle neurons rather than neurons with neurofibrillary tangles in AD [66], and in dopaminergic neurons with diffused α-synuclein in PD postmortem brains [67]. Although we focused on the later time point (21 dpi), when PrP$^{Sc}$ accumulation was observed in neurons, it is of interest if acute toxicity of PrP$^{Sc}$ [15, 41] or an immediate response of neurons to prions stimulates the activation of PERK and its downstream pathways despite significantly lower level of PrP$^{Sc}$ accumulation than later time periods. Alternatively, it is reported that PrP$^{Sc}$ causes acute toxicity in excitatory synapses through glutamate receptor and p38 MAPK activation, which is not presumably related to UPR [68]. Thus, analyses of earlier time points are also required to evaluate neuron-autonomous responses to prion infection.

We described the induction of ER stress as genuine neuronal responses to prion propagation; however, it did not proceed to a pro-apoptotic phase or synaptic losses autonomously in neurons *ex vivo*. Our findings suggest the presence of non-neuronal factor(s) in the mechanism of neurodegeneration in prion diseases, which may provide clues for therapeutics to prion diseases and other misfolding disorders.

## Supporting information

**S1 Fig. Percentages of NeuN-positive neurons and GFAP-positive astrocytes in primary cultures used in this study.** Primary neuronal cultures were treated with antimitotic AraC at 0.25 μM from 4 days in vitro (div) to 7 div in the first week. Cells were immunostained for NeuN and GFAP to identify neurons and astrocytes in the cultures at 21 dpi (corresponds to 28 div). MetaMorph software was used to set a threshold of fluorescent intensity of each cell marker. Cells which were not immunostained for NeuN and GFAP, and have shrunk and condensed nuclei (DAPI-positive area $\leq 80$ μm$^2$) were considered as putative dead cells and removed from counting. Images show representative CxNs and ThNs cultured under each condition. Arrows, filled arrowheads, and open arrowheads indicate NeuN-positive neurons, GFAP-positive astrocytes, and unidentified cells (NeuN- and GFAP-negative cells), respectively. The table shows percentages of each cell type in the cultures. Bar, 50 μm.
(TIF)

**S2 Fig. Representative immunofluorescence images.** Representative immunofluorescence images of (A) CxNs and (B) ThNs used for image analyses shown in Fig 2. From the left, panels show merged images, subsets of NeuN and DAPI, and binary images of MAP2 staining. (C & D) Unnormalized values of neuronal cell density and neurite density for (C) CxNs and (D) ThNs measured by MetaMorph software. Cell density and neurite density were defined as the number of NeuN-positive nuclei per 0.01 mm2 and the surface coverage (%) by MAP2-positive neurites, respectively. Bar graphs show mean ± SD of 9 replicates from 3 independent experiments. Bars = 50 μm.
(TIF)

**S3 Fig. Ratios of phosphorylated PERK (p-PERK) to total PERK (t-PERK) and phosphorylated eIF2α (p-eIF2α) to total eIF2α (t-eIF2α).** Phosphorylation levels of PERK and eIF2α were assessed as ratio between the phosphorylated and total protein signals quantified by immunoblotting described in Fig 3. Ratios of p-PERK/t-PERK (top) and p-eIF2α/ t-eIF2α (bottom) are indicated in Tables (mean ± SD). Although differences were not statistically significant, p-PERK/t-PERK ratios tended to be higher in prion-infected CxNs and ThNs than mock-infected CxNs and ThNs. In contrast to ratios of p-PERK/t-PERK, those of p-eIF2α/t-

eIF2α did not differ between prion-infected and mock-infected primary neurons.
(TIF)

**S4 Fig. Immunostaining for p-PERK (Thr980) using a rabbit monoclonal antibody (16F8) in tunicamycin-treated Neuro2a cells.** Immunofluorescence staining for p-PERK (green) with counter staining for nuclei (DAPI, blue) was carried out using Neuro2a cell treated with tunicamycin at indicated concentrations for 12 hrs. Images are shown as maximum intensity projection created from z-series stacks of confocal images. Fluorescent granular signals stained by this antibody appeared more in tunicamycin-treated Neuro2a cells (5.0 μg/ml) than in untreated Neuro2a cells, indicating that the specificity of reaction of this antibody in IFA. Bars, 10 μm.
(TIF)

**S5 Fig. Phosphorylation of PERK in PrP$^{Sc}$-positive thalamic neurons.** (A) Multiple immunofluorescence staining for p-PERK and PrP$^{Sc}$ with counter staining for MAP2 and nuclei (DAPI) in ThNs at 21 dpi. PrP$^{Sc}$ was stained with mAb 8D5. Figures are shown as maximum intensity projection images. Arrows indicate a PrP$^{Sc}$++ neuron. Bars, 50 μm. (B, C) Quantification of p-PERK and PrP$^{Sc}$ in individual neurons by 3D-image analysis using Imaris software. Each dot in the scatter diagram (B) represents an individual thalamic neuron although some neurons were spotted at the same position (r, correlation coefficient). In (C), Obihiro-infected ThN was analyzed as a whole (Obihiro) and as three subpopulations classified by the frequency of PrP$^{Sc}$ signals at the soma. The definition of the three subpopulations and the number of cells used for this analysis are as follows: PrP$^{Sc}$_low ($< 4$ PrP$^{Sc}$ signals/cell, n = 49), PrP$^{Sc}$+ ($\geq 4$ PrP$^{Sc}$ signals/cell, n = 26, including PrP$^{Sc}$++), and PrP$^{Sc}$++ ($\geq 13$ PrP$^{Sc}$ signals/cell, n = 12). The number of cells in mock-infected ThN and Obihiro-infected ThN used for the analysis are n = 49 and n = 75, respectively. Differences between the groups were analyzed by Steel-Dwass's multiple comparison tests. $^{*}$, p $< 0.05$
(TIF)

**S1 Raw images.**
(PDF)

## Acknowledgments

We thank Dr. Roth B. for pAAV-hSyn-EGFP, and to Dr. Gradinaru V. for pUCmini-iCAP-PHP.eB. We also thank Zensho Co., Ltd, for the BSL3 facility.

## Author Contributions

**Conceptualization:** Motohiro Horiuchi.

**Funding acquisition:** Motohiro Horiuchi.

**Investigation:** Misaki Tanaka, Takeshi Yamasaki, Rie Hasebe, Akio Suzuki.

**Supervision:** Motohiro Horiuchi.

**Writing – original draft:** Misaki Tanaka, Motohiro Horiuchi.

**Writing – review & editing:** Motohiro Horiuchi.

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
