## [Decision Letter · Decision Letter 0]

18 Mar 2020

PONE-D-20-03488

Enhanced phosphorylation of PERK in primary cultured neurons as an autonomous neuronal response to prion infection

PLOS ONE

Dear Dr. Horiuchi,

Thank you for submitting your manuscript to PLOS ONE. After careful consideration, we feel that it has merit but does not fully meet PLOS ONE’s publication criteria as it currently stands. Therefore, we invite you to submit a revised version of the manuscript that addresses the points raised during the review process.

It is important that all the technical points are addressed, and your results are discussed in light of previous studies investigating the role of the UPR in prion pathogenesis.

We would appreciate receiving your revised manuscript by May 02 2020 11:59PM. To enhance the reproducibility of your results, we recommend that if applicable you deposit your laboratory protocols in protocols.io, where a protocol can be assigned its own identifier (DOI) such that it can be cited independently in the future. For instructions see: http://journals.plos.org/plosone/s/submission-guidelines#loc-laboratory-protocols

We look forward to receiving your revised manuscript.

Kind regards,

Roberto Chiesa

Academic Editor

PLOS ONE

Journal Requirements:

Reviewers' comments:

Reviewer's Responses to Questions

**Comments to the Author**

1. Is the manuscript technically sound, and do the data support the conclusions?

Reviewer #1: Yes

Reviewer #2: Partly

Reviewer #3: Yes

2. Has the statistical analysis been performed appropriately and rigorously? 

Reviewer #1: Yes

Reviewer #2: Yes

Reviewer #3: Yes

3. Have the authors made all data underlying the findings in their manuscript fully available?

Reviewer #1: Yes

Reviewer #2: Yes

Reviewer #3: Yes

4. Is the manuscript presented in an intelligible fashion and written in standard English?

Reviewer #1: Yes

Reviewer #2: Yes

Reviewer #3: Yes

5. Review Comments to the Author

Reviewer #1: In this manuscript, Tanaka and colleagues analysed the autonomous neuronal response to prion infection. They used neuron-enriched primary cultures of cortical, which have been previously characterised (Tanaka M. et al., 2016), and thalamic neurons. They infected neurons with the mouse-adapted scrapie Chandler and Obihiro strains.

They found that both cortical and thalamic cultures successfully propagate prions Chandler and Obihiro strains. Despite the continuous increasing of PrPSc levels in neurons over time, no obvious and progressive neuronal loss or neurite alteration were observed in prion-infected neurons compared to mock-infected neurons. The authors measured neuron density as NeuN positive nuclei per 0.01 mm2. I wonder what the percentage of NeuN positive cells on the total DAPI positive cells is, since it seems that only few cells are positive to NeuN in fig S1.

Phosphorylation of PERK was enhanced in prion-infected cortical neurons as shown by immunoblot analysis. As normally phosphorylation level of PERK and eIF2alpha is expressed as ratio between the phosphorylated and total protein signal, I suggest to express the data in figure 3 as ratio p-PERK/PERK and p-eIF2α/eIF2α. The authors performed also a finer IF analysis of individual p-PERK in PrPSc-positive cortical neurons at 21 dpi, that demonstrated a positive correlation between the number of PrPSc and p-PERK granular stains.

Interestingly, despite PrPSc production caused PERK phosphorylation, neither translational repression of synaptic protein synthesis nor activation of downstream effectors of PERK-eIf2α pathway were observed. These results indicate that PrPSc production induces ER stress in cortical neurons in cell autonomous manner, but this is not sufficient to fully activate the UPR leading to neurodegeneration. The authors suggest the involvement of neuronal non autonomous factors in the mechanisms of neurodegeneration in prion disease. This is in agreement with the recent finding that dysregulation of PERK signaling in astrocytes induces a specific reactivity state causing non-cell autonomous pathogenic mechanism in prion neurodegeneration (Smith H.L. et al., 2020).

All experiments are well done and have the appropriate controls.

This work is well written and clear.

Typing errors:

line 28 alternations (alterations)

line 109 autonomaous neurnal (autonomous neuronal)

In line 283 is SE or SD?

The percentage of PrPSc positive ThN is 54.8% (line 274) or 52.8% (line 422)?

line 503 *p<0.001 (***p<0.001)

Reviewer #2: In this paper, the authors investigate the role of the UPR in prion infection using cultured cortical and thalamic neurons in combination with Western blotting and immunostaining. They conclude that, although prion infection is correlated with increased PERK phosphorylation, there was no evidence of subsequent translational repression of synaptic protein synthesis or activation of downstream steps of the UPR pathway. There has been previous work on the role of the UPR in prion infection, most notably by Mallucci and colleagues, who showed that prion infection in vivo fully activates the PERK branch of the UPR, and that genetic or pharmacological manipulation this branch mitigates prion disease symptoms and pathology in mice. In contract, Fang et al. (PLoS Pathogens 14(9): e1007283, 2018) showed that UPR inhibitors had no effect on the acute synaptotoxic effects of PrPSc on cultured hippocampal neurons.

This paper aims to investigate an important question: the molecular and cellular mechanisms by which prions damage neurons. However, there are both conceptual as well as technical flaws, which detract from the impact of the paper. As a result, this study does not make a major contribution to advancing knowledge about the role of the UPR in prion diseases.

Major criticisms:

1. Why does prion infection activate only PERK phosphorylation, but not the downstream steps of the pathway? The authors do not offer much speculation or experimental evidence on this issue.

2. Why does PERK phosphorylation increase only in cortical and not thalamic neurons? This calls into question the generality of the results.

3. The authors do not do a very good job reconciling their results with those of Mallucci or Fang et al. (2018).

Technical criticisms:

4. What is the proportion of neurons in culture? When was AraC added to the cultures? These primary cultures probably contain a mixture of cell types, and non-neuronal cells may account for up to 30% of total cells (Ref. #17). Therefore, immunoblot analysis for protein levels (Fig. 3) will represent the responses of multiple cell types, and not just neurons. Even though the authors normalized their Western blot data using βIII tubulin, this will only equalize the proportion of neuronally-derived protein in each sample, but not remove the effect of non-neuronal cells. This problem is compounded in the normalization for the pre- and post-synaptic markers (Fig. 4), since the proportion of neuronal vs. non-neuronal cells is likely changing over time even in the absence of prion infection.

5. The authors used only the phospho-PERK (Thr980) (16F8) antibody, but this product is recommend only for Western blots. This raises a question about the specificity of the IF staining using this antibody.

Reviewer #3: The manuscript by M. Tanaka and colleagues describes enhanced phosphorylation of PERK in primary neuronal cultures as a response to prion infection. The authors performed a very careful and well controlled analysis and studied the molecular mechanisms of neurodegeneration in prion disease in primary cells. The authors infected neuron-enriched primary cultures (cortical and thalamic) with two strains of prions and tested them for alterations in neuronal and neurite integrity in immunoblot, RT-PCR and immunofluorescence assays. Only minor indications for ER stress were found, like increased phosphorylation of PERK, and there was no translation into full activation of an unfolded protein response (UPR) in these infected primary neurons. The major conclusion from these findings was that non-neuronal factors are needed for full execution of UPR in prion-infected neurons, most likely coming from astrocytes. Evidence for this conclusion is very compelling, although the authors did not discuss in the manuscript how pure their neuronal cultures at the end really are. They provide references to earlier publications, but it is necessary to provide this data or at least discuss them. Taken together, the authors provide important data which are novel, significant and which have the potential to exert an impact to the field. The manuscript is well done, experiments are clearly described and well controlled, and conclusions are fully justified by the experimental data.

Minor points:

1) Page 8/Immunoblotting: Duration of PK digestion should be indicated.

2) Page 8/IF and page 18: mAb 8D5 should also be described in Materials and Methods, with clearly mentioning that no GndSCN step is done.

3) Fig. 1A/B: It would help to indicate in the figure and/or charts that A shows CxN and B ThN neurons.

4) Fig 1: The rationale of relating quantification of PrPSc to day 7 p.i. should be provided.

6. PLOS authors have the option to publish the peer review history of their article (what does this mean?). If published, this will include your full peer review and any attached files.

Reviewer #1: No

Reviewer #2: No

Reviewer #3: No

---

## [Author Response · Author response to Decision Letter 0]

4 May 2020

Please see Cover Letter and Response to Reviewers

---

## [Decision Letter · Decision Letter 1]

20 May 2020

Enhanced phosphorylation of PERK in primary cultured neurons as an autonomous neuronal response to prion infection

PONE-D-20-03488R1

Dear Dr. Horiuchi,

We are pleased to inform you that your manuscript has been judged scientifically suitable for publication and will be formally accepted for publication once it complies with all outstanding technical requirements.

With kind regards,

Roberto Chiesa

Academic Editor

PLOS ONE

Additional Editor Comments (optional):

Please amend the following in the final version of the manuscript:

1) Page 16, line 369-371: "In contrast to ratios of p-PERK/total PERK, those of p-eIF2α/ total eIF2α did not differ between prion-infected and mock-infected primary neurons. "

"In contrast to ratios of p-PERK/total PERK..." should be removed since in the previous sentence it is stated that there is no statistically significant differences in p-PERK/total PERK ratios.

2) Page 25, line 591: "...no differences were observed in the densities PSD95-..." should read "...no differences were observed in the densities **of** PSD95-..."

3) page 26, line 14: please replace "highlight" with "suggest".

Reviewers' comments:

Reviewer's Responses to Questions

**Comments to the Author**

1. If the authors have adequately addressed your comments raised in a previous round of review and you feel that this manuscript is now acceptable for publication, you may indicate that here to bypass the “Comments to the Author” section, enter your conflict of interest statement in the “Confidential to Editor” section, and submit your "Accept" recommendation.

Reviewer #1: All comments have been addressed

Reviewer #3: All comments have been addressed

2. Is the manuscript technically sound, and do the data support the conclusions?

Reviewer #1: Yes

Reviewer #3: Yes

3. Has the statistical analysis been performed appropriately and rigorously? 

Reviewer #1: Yes

Reviewer #3: Yes

4. Have the authors made all data underlying the findings in their manuscript fully available?

Reviewer #1: Yes

Reviewer #3: Yes

5. Is the manuscript presented in an intelligible fashion and written in standard English?

Reviewer #1: Yes

Reviewer #3: Yes

6. Review Comments to the Author

Reviewer #1: (No Response)

Reviewer #3: (No Response)

7. PLOS authors have the option to publish the peer review history of their article (what does this mean?). If published, this will include your full peer review and any attached files.

Reviewer #1: No

Reviewer #3: Yes: Hermann M. Schatzl

---

## [Editor Report · Acceptance letter]

22 May 2020

PONE-D-20-03488R1 

Enhanced phosphorylation of PERK in primary cultured neurons as an autonomous neuronal response to prion infection 

Dear Dr. Horiuchi:

I am pleased to inform you that your manuscript has been deemed suitable for publication in PLOS ONE. Congratulations! Your manuscript is now with our production department. 

With kind regards,

on behalf of

Dr. Roberto Chiesa 

Academic Editor

PLOS ONE